# Self-Management Support for Cancer Survivors: A Descriptive Evaluation of the Symptom Navi Training from the Perspective of Health Care Professionals

**DOI:** 10.3390/curroncol32060326

**Published:** 2025-06-02

**Authors:** Marika Bana, Selma Riedo, Karin Ribi

**Affiliations:** 1School of Health Sciences, University of Applied Science and Arts Western Switzerland, HES-SO FR, 1700 Fribourg, Switzerland; selma.riedo@hefr.ch; 2Careum School of Health, Part of the Kalaidos University of Applied Sciences Zurich, 8006 Zurich, Switzerland; karin.ribi@careum-hochschule.ch

**Keywords:** self-management support, cancer survivorship, oncology nursing, work context, Work-SoC scale, Explorative Factor Analysis, behaviour change

## Abstract

The Symptom Navi Program (SNP) is a self-management support (SMS) intervention for people with cancer. It consists of self-management supportive leaflets, educational conversations, and two standardized training sessions. A descriptive quality evaluation method was used to evaluate SNP implementation across 14 cancer services from 2021 to 2024. We evaluated training content, methods, and participants’ confidence to use SMS in their clinical routine. Nurses, social workers, and psychologists completed ad hoc closed and open-ended questions after each training. The Work Sense of Coherence (Work-SoC) scale was used to elicit participants’ self-reported perceptions of their work context at cancer services. A series of descriptive analyses were conducted on the Work-SoC scale, the training content, and the methods. In addition, training-specific questions and predefined hypotheses were correlated. Thematic analysis was employed to examine open-ended questions. The SNP training content and methods largely met participants’ needs. Participants’ confidence in applying educational conversations decreased over time. The findings suggest a robust correlation between the application of educational conversations in daily routines and the participants’ perceptions regarding the comprehensibility and manageability of their work situations. Future research focusing on the implementation of SMS in clinical practice should examine the work context.

## 1. Introduction

The number of people living with cancer continues to increase worldwide [1] and in Switzerland [2]. The availability of modern anticancer treatments is evolving rapidly, and while they are effective in terms of prolonging progression-free survival or overall survival, they can be associated with a range of acute and long-term side effects that negatively affect quality of life [3]. In addition, treatment has shifted from inpatient to outpatient and home care. Patients leaving the cancer clinic are required to monitor, report, and manage their disease- and treatment-related symptoms and to deal with emotional challenges and psychosocial consequences that can affect everyday life [4]. Therefore, the Global Partners for Self-Management in Cancer, an international collaboration of researchers, have identified six actionable priorities to better integrate self-management support (SMS) as part of high-quality, person-centred cancer care: (1) to prepare patients, survivors, and caregivers for active involvement in care, (2) to embed SMS into everyday practice and care pathways, (3) to train health care providers with knowledge and skills for providing SMS, (4) to foster accountability for SMS as a performance metric in value-based care, (5) to advance evidence on the effectiveness of self-management and SMS in cancer populations, and (6) to expand reach and access to self-management programs across care sectors [4]. Self-management refers to ‘an individual’s ability to cope with symptoms, treatment, physical and psychosocial consequences, and lifestyle changes associated with living with a chronic condition’ [5] (p. 2).

There is a growing body of evidence on the different approaches to SMS content, type, theoretical underpinning, and impact that have been introduced and tested in people with cancer, with varying effects on outcomes [6,7,8,9]. Key elements of self-management in cancer include problem solving, decision making, self-monitoring and adjusting behaviour, goal setting and action planning, partnering with healthcare professionals, and maintaining health by reducing risks [4]. However, oncology healthcare professionals typically have little understanding of these key elements of self-management [10]. Training healthcare professionals with the knowledge and skills to support patients in self-management is therefore a crucial step in the transformation of care from reactive to proactive [4].

In Switzerland, the Symptom Navi Program (SNP) was developed more than a decade ago to address the lack of standardized approaches to support the self-management of cancer patients in the early stages of treatment [11]. The SNP was developed in collaboration with healthcare professionals and patients diagnosed with cancer for oncology nurses to complement usual care. It consists of written information leaflets (SN leaflets), nurse-led semistructured educational consultations using these leaflets, and standardized SNP implementation training based on two training sessions [12]. Both training sessions focus on facilitating behaviour change to apply a coaching approach to SMS. Therefore, we based the SNP training on the Capability Opportunity Motivational—Behaviour (COM-B) model [13], which is described in the SNP training manual. The COM-B model emphasizes that changes in nursing practice depend on knowledge and skills (capabilities), analytical choices (motivation), and contextual factors (opportunities) that enable the desired behaviour. In addition to the SNP components, we developed an SNP manual for health care providers describing the practical application of the SNP to facilitate long-term implementation in cancer services. In this manual, we outlined the content of the SNP training and detailed the theoretical underpinnings of SMS, such as self-efficacy [14] and the Theory of Self-Management [15]. Further details of the SNP training procedures have been published elsewhere [11,12].The feasibility of implementing the SNP in clinical practice and the preliminary effectiveness of the SNP on patient outcomes have been tested in a pilot study [16].

Face-to-face SMS delivered by nurses requires personal and institutional resources. Evidence suggests that the work environment should be considered as an important factor for successful implementation [10,17,18]. The SNP pilot study has collected initial data on nurses’ perceptions of the acceptability of and confidence in the use of the SN leaflets in educational conversations. It also explored nurses’ confidence in implementing the SNP in their daily clinical practice in relation to their perceptions of their current work situation based on the concept of work-related sense of coherence [11]. Work-related Sense of Coherence (Work-SOC) is predicated on the salutogenetic approach, which places emphasis on the genesis of health rather than risks [19] and refers to the extent to which individuals perceive their work situation as comprehensible, meaningful, and manageable [20]. The Work-SoC scale complements the COM-B model and can facilitate interpretation on training evaluation. Data from fourteen nurses showed that their confidence to use educational conversations was positively correlated with their estimated overall Work-SoC score, suggesting that the work situation may be a critical factor in implementing the SNP in their clinical daily routine [11].

Fidelity to training content and intervention procedures is key for long term implementation of a new intervention. Based on the Medical Research Council guideline [21], supporting self-management is considered as a complex intervention: educational conversations are always tailored to an individual situation (cancer diagnosis, disease stage, treatment regimen, social status) and follow a standardized but flexible coaching approach [11].

Following the pilot testing of the SNP, the program has been disseminated to fourteen more institutions, and other professionals than oncology nurses have been trained. Health Promotion Switzerland (www.gesundheitsfoerderung.ch) supported a project to disseminate the SNP to more remote oncology services, including other healthcare professionals such as social workers, psychologists, and nurses working in the home setting [22]. Scaling-up the use of SNP across different oncology services and health care professionals requires ongoing quality assessment. Therefore, we extended the evaluation of each SNP training using the same evaluation tools as in the pilot study.

The objective of this program evaluation involving fourteen cancer services is threefold: first, to evaluate the extent to which the SNP training content and methods meet the needs of participants in their work context to complement the data from the pilot study; second, to learn from the feedback of training participants on how to improve the implementation of the SNP; and third, to investigate how the work situation of the trainee may affect the successful implementation of the SNP.

## 2. Materials and Methods

### 2.1. SNP Training Evaluation Design

This descriptive program evaluation assessed participants perceptions regarding the SNP training content and methods, their confidence to use educational conversations in daily routine, and their perception of their individual work situation at two time points. Both quantitative and qualitative methodologies were used. We based the evaluation methods on the Medical Research Council guidelines for the development and evaluation of complex interventions [21].

We contacted oncology services to participate in the project supported by Health Promotion Switzerland [22] and trained oncology services that expressed interest in implementing the SNP since the end of the SNP pilot study. We used ad hoc closed and open questions to capture training content and methods and the Work-related Sense of Coherence (Work-SoC) scale [20,23] to evaluate work-related factors considered to potentially impact the feasibility of implementing the SNP at different oncological services in Switzerland. We included healthcare professionals who supported SMS for people diagnosed with cancer under treatment, after cancer treatment is completed, and in palliative care settings. SNP training took place between 2021 and 2024.

Evaluation questions were:How do the SNP training content and methods meet participants’ requirements in their clinical context?What can we learn from training participants’ feedback to facilitate the implementation of the SNP?How do the training participants estimate their work situation as assessed by the Work-SoC scale?

The SNP pilot study showed that organizing SNP training sessions was time-consuming for oncology services. Training the entire team at multiple facilities presented a considerable challenge. To address these difficulties, we chose a pragmatic quality evaluation to facilitate organization and participation to the training sessions. Therefore, we continued the standardized program quality evaluation without assessing any sensitive information such as age, gender, or education; nor did we assess any health-related information. Nevertheless, the training evaluation was conducted in accordance with the Declaration of Helsinki [24], the principles of good clinical practice, and the Swiss Human Research Ordinance [25]. Training participants completed anonymous paper-based questionnaires after each training session, either in German or in French. Participants were free to decline answering any questions.

### 2.2. Participants and SNP Training Procedures

Cancer services in German-, French-, and Italian-speaking regions of Switzerland have implemented the SNP. The services under consideration were located in the following facilities: tertiary or regional hospitals; oncological outpatient clinics; oncological medical practices; or cancer league services. Registered nurses, psychologists, and social workers who attended at least one SNP training session were included. The participating oncology services determined their team members to attend the training sessions. The number of participants who could join the training was determined by local and organizational resources. In certain instances, this involved the entire team or designated team members.

We conducted the training in two sessions in German or French: the initial training, which occurred at the onset of SNP implementation at a given cancer service (i.e., the baseline), and the follow-up training, which occurred a minimum of eight weeks after the initial training [11]. The initial four-hour training aimed to familiarize participants with the SNP, to reiterate motivational interviewing, and to introduce a coaching approach based on the 5 A’s (assess, advise, agree, assist, arrange) [26]. Motivational interviewing [27] is a well-integrated approach that supports behaviour change and is part of the basic nursing training in Switzerland. The 5 A’s approach is recommended for the structuring of SMS interventions [26,28]. The initial training established the foundation for conducting educational conversations with SN leaflets by emphasizing participants’ capabilities and motivation to use educational conversations. A variety of instructional methods were used to introduce a coaching SMS approach with the 5 A’s and motivational interviewing. These methods included oral presentations (expert inputs) and interactive group exercises (roleplays and team discussions). The objective of the two-hour follow-up training was to strengthen long-term implementation in collaboration with the healthcare team. This training session focused on the practical application of SMS in daily routines (opportunities) and addressed any queries regarding the initial training. Both training sessions were conducted by nurses with expertise in oncological care who had obtained a minimum of a master’s degree. Trained centres received a manual (available in French and German) that provided details on the theoretical underpinning of SMS, explained how to foster educational conversations, and accentuated inputs from the training. This manual was a supplementary tool to be consulted as needed.

### 2.3. Measures

#### 2.3.1. Training-Specific Questions

Five questions were developed to capture the objectives of initial and follow-up training. Participants rated the questions related to training content and methods, as well as questions related to their confidence in applying SMS, on a 7-point Likert-like scale (to what extent a statement was true: 1 = ‘not at all’ to 7 = ‘very much’). Two open-ended questions were asked to obtain both positive and negative feedback on the training. Respondents were invited to offer brief comments and/or key words in response to these questions.

#### 2.3.2. Work-SoC Scale

The Work-SoC scale is a 9-item instrument using a 7-point Likert scale assessing the three subdimensions *comprehensibility*, *meaningfulness*, and *manageability*. Each questionnaire item is designed to elicit a response between opposing answers, such as, for example, clear–unclear. The Work-SoC scale was originally tested in a cross-sectional and longitudinal study including 3412 participants from different work contexts. In this study, the scale had a Cronbach alpha of 0.83 and three subdimensions: *comprehensibility* (4 items), *meaningfulness* (3 items), and *manageability* (2 items). A recent study evaluating team resilience at work in the oncological context showed a lower Cronbach alpha of 0.66 yet confirmed the factor loading of the items to the three subdimensions [29]. Cronbach’s alpha in this evaluation was 0.64, and explorative factor analysis (EFA) of the Work-SoC scale confirmed a three factor-structure with three items loading on each factor (Appendix A).

### 2.4. Statistical Analysis

First, we performed an exploratory factor analysis (EFA) using principal axis factoring (PAF) and the varimax rotation to uncover the latent constructs of the Work-SoC items and compare the factor loading of our data to the three subdimensions *comprehensibility*, *meaningfulness*, and *manageability*. We also examined the scree plot to determine the optimal number of factors to retain and checked overall model adequacy using Bartlett’s test and the KMO measure (for results see the Appendix A). Second, we analysed the training-specific questions and the Work-SoC scale items descriptively (mean, standard deviation (SD), minimum/maximum). For the Work-SoC scale, we also analysed the difference between initial and follow-up training for each item to observe potential changes in perceptions over time. Third, we tested for correlations between training-specific questions and predefined hypotheses.

Preliminary results of the SNP pilot study, which included a small sample size of 14 nurses, showed that the overall Work-SoC scale was positively associated with participants’ confidence to implement educational conversations in daily routines (rπ = 0.47, *p* = 0.04) [11]. These results suggest a potential association between participants’ perceptions of the meaningfulness of the work situation and their estimates regarding the use of SMS after the initial training. Furthermore, we hypothesized that the participants’ perceptions of the comprehensibility and manageability of their work situation would be associated with the continuation of SMS after the follow-up training. To gather more insight into which subdimensions related to the work context may impact providing SMS in daily routines, we hypothesized:That the Work-SoC subdimension *meaningfulness* was associated with the statements “I have the confidence to use educational conversations in daily routines” and “I have the confidence to use motivational interviewing during educational conversations” after initial training (baseline assessment);That the subdimensions *comprehensibility* and *manageability* were associated with the statements “I feel empowered to apply educational conversations in my work context” and “I feel confident to integrate educational conversations in daily routines” after the follow-up training.

The statistical software R version 4.3.0 with the package *psych* was used. Statistical significance was set at *p* ˂ 0.05.

### 2.5. Thematic Anaysis

To analyse the narrative feedback (short statements in response to open questions) on the training, we used Braun and Clark [30] thematic analysis, which is based on six phases: (1) familiarizing with data, (2) generating initial codes, (3) searching themes, (4) reviewing themes, (5) defining and naming themes, and (6) producing a report. The first three phases were conducted using an inductive approach with the NVivo software program version 14.24.3 (7). Subsequently, Phases 4 and 5 were conducted deductively, using the three Work-SoC subdimensions *comprehensibility*, *meaningfulness*, and *manageability*. Two collaborators independently completed the first three phases and discussed the codes and themes until consensus was reached. The first author was responsible for conducting phases 4 and 5 for interpreting the quantitative results. These results were subsequently discussed by the author team.

Quantitative and qualitative analyses were carried out consecutively. The results of the thematic analysis were used to reflect on the interpretation of the quantitative results. 

## 3. Results

The training was conducted from February 2021 to June 2024 at fourteen different cancer services. We trained health care professionals at nine cancer services in tertiary or regional hospitals, two outpatient cancer centres, two cancer leagues, and a mobile outpatient palliative care team. At the initial training, 134 nurses, 6 social workers, and 1 psychologist participated; 43 nurses and 3 social workers participated at the follow-up training. The number of participants at each cancer service varied between 6 and 24 participants corresponding to institution size (see Figure 1). Training hours lasted between 3 and 4 h for initial and between 1.5 and 2 h for follow-up training.

Healthcare professionals from ten of the fourteen services participated in both training sessions. Because of scheduling constraints, it was not possible to arrange the follow-up training to meet the required time interval for two centres. An oncology ambulatory and a regional hospital decided against organizing a follow-up training. At the conclusion of the initial training, all participants completed the questionnaires immediately. Two centres dispatched the questionnaires later via email; however, four participants did not respond. After the follow-up session, all participants at nine centres promptly completed the questionnaires immediately after the training. One tertiary hospital did not return the questionnaires via email. One of the follow-up trainings was conducted online, while all the other initial and follow-up trainings were held in-person at the oncology service.

### 3.1. Descriptive Analysis of the SNP Training

Initial and follow-up training content largely met participants’ expectations (see Table 1 and Table 2). After the initial training, participants felt confident to apply educational conversations (Mean 5.5, SD 1.3) and to use motivational interviewing (Mean 5.2, SD 1.4) (Table 1). Discussing case studies seemed to be less helpful for the learning process for some of the participants.

Considerably fewer healthcare professionals participated in the follow-up training. Four institutions did not participate in the follow-up training or have not yet participated. This clarifies the substantial decrease in the number of participants, from 150 to 46. Participants confirmed that they could ask questions and got satisfactory answers. The majority felt empowered to apply educational conversations and felt confident to explain SN leaflets. However, participants showed a medium level of confidence to actually apply educational conversations in their daily routines (Mean 4.8, SD 1.7) after the follow-up training (Table 2).

### 3.2. Work-Related Sense of Coherence (Work-SoC)

Descriptive results on the Work-SoC scale are summarized in Table 3. Overall, participants perceived their work situation as meaningful, with rather low scores indicating a better rating for all items related to this subdimension. Items related to the *comprehensibility* and *manageability* subdimensions were rated with greater variability, with higher mean scores suggesting that these two subdimensions were considered challenging. No statistically significant difference was observed between baseline and follow-up Work-SoC scores except for manageable work situation (*p* = 0.035).

### 3.3. Work-SoC Hypothesis Testing

Hypothesis 1 was not supported. Therefore, the perceived meaningfulness of an individual’s work situation does not appear to be associated with the use of educational conversations (ρ\rhoρ: −0.0064, *p* = 0.9467) or the use of motivational interviewing (ρ\rhoρ): 0.0055, *p* = 0.9541) following the initial SNP training.

Hypothesis 2 received substantial support, as participants’ perception of the *manageability* of their work situation (i.e., work situations that are easily influenced, controllable, and predictable) were significantly associated with their sense of empowerment and their confidence in implementing educational conversations following the follow-up training. Nevertheless, participants’ estimation of the *comprehensibility* of the work situation (i.e., how structured, clear, and manageable the work situation is) was not significantly associated with the degree to which participants felt empowered to apply educational conversations after the follow-up training (see Table 4). The negative correlation suggests that higher Work-SoC subdimension scores (i.e., a lower sense of coherence) are associated with a decrease in feeling empowered or confident to apply educational conversations.

### 3.4. Qualitative Evaluation of the SNP Training

The narrative statements from both trainings were divided into rather positive (n = 192) and rather negative statements (n = 71). Initial coding revealed that positive and negative feedback was related to training methods and the program’s transferability to daily routines and that only positive statements were reported on team development and training atmosphere. Themes related to methods included *SNP quality*, *transferability*, *comprehensibility*, *discussions*, and *training atmosphere*. Positive *team strategy* and inappropriate *training schedule* were identified as complementary themes.

Positive statements after the initial training (n = 152) were related to program quality (evidence-based self-management recommendations, underlying theoretical framework), an inspiring learning atmosphere, and the opinion that educational conversations with the SN leaflets are feasible for use in clinical routine. The training methods were in general well received by participants. Positive statements indicated that the initial training content and methods largely met participants’ needs and expectations. However, rather negative statements after the initial training (n = 61) concerned some training methods (too much theory) or doubts caused by a feeling of insufficient training for motivational interviewing techniques. These statements suggest that theoretical inputs and motivational interviewing techniques might be challenging for some participants.

Rather positive about the follow-up training (n = 40) was that some centres experienced team-building spirit. This was due to an effect of common goals for SMS at their service. Ten negative statements after follow-up training addressed issues related to the timing of the training and unmet learning needs. Because of part-time engagement or annual leave, not all participants were able to apply educational conversations between initial and follow-up training. As a result, they considered the follow-up training to be too early. Some participants in the follow-up training expected to dive into SMS techniques rather than focus on opportunities related to long-term implementation. After the follow-up training, participants also raised more scepsis regarding educational conversations and time constraints due to very busy routines. A selection of quotes related to initial and follow-up training feedback are illustrated in Table 5.

## 4. Discussion

This descriptive program evaluation had three objectives: (1) to evaluate SNP training content and methods, (2) to describe what we can learn from participant feedback to facilitate implementation, and (3) to describe how participants estimate their work situation.

In general, the evaluation of the SNP showed that the content and the delivery methods of the SNP training met the needs of the participants to successfully use the SN leaflets in educational conversations in their clinical context. After the initial training, nurses, social workers, and psychologists embraced the SNP training and felt confident in conducting educational conversations. Participants appreciated the training methods, including expert input, roleplaying to practice educational conversations, and team discussions. An external evaluation of the SNP confirmed that healthcare professionals were satisfied with the training sessions. They emphasized the importance of SMS for cancer patients and that the SN leaflets were a valuable resource for SMS [22]. However, some participants raised concerns about the feasibility of educational conversations and SMS in busy outpatient services related to time and staff resources at their centre. Concerns about time and staff resources in outpatient oncology settings have been frequently reported in other studies on self-management interventions [10,31,32]. Although standardized self-management interventions could make it easier for health care professionals to integrate educational conversations into daily routines by providing clear structure and guidance, implementation in busy oncology clinics still seems to be a challenge. According to the International Society of Nurses in Cancer Care (ISNCC) framework by Chan and colleagues, cancer nurses need the following skills and competencies to provide SMS to cancer patients: (1) person-centred and motivational interviewing communication skills, (2) whole-person assessment of self-management support needs and capacity, (3) health promotion theories and interventions, (4) coaching for behaviour change tailored to the individual’s phase in the cancer continuum, (5) monitoring and evolution of self-management behaviours and health outcomes, and (6) quality improvement for integration SMS in cancer care [33]. The SNP training sessions covered the majority of these competencies; however, they did not explicitly integrate techniques for assessing SMS needs and capabilities, and they did not integrate the evaluation of quality improvement or the monitoring of self-management behaviour. These missing elements may be the reason why we observed a decrease in confidence in using educational conversations at the follow-up training.

Participants’ feedback offers several valuable lessons. Following the initial training, they felt confident to use the SN leaflets and apply SMS. The positive adoption of the SNP appeared to decrease after the follow-up training at certain oncology services. As we did not assess personal information related to participants, we were unable to distinguish between the different healthcare professions. However, discussions at follow-up trainings with participants revealed that social workers and psychologists lacked confidence in their ability to apply SMS. If necessary, they preferred to refer clients to an oncology team. An unfavourable work situation may explain why confidence was not maintained after the initial training. Certain statements indicate that some participants perceived the work situation as not supportive to implementing SMS. Additionally, some participants noted a lack of comprehensive training in motivational interviewing and expressed a desire for further elaboration of SMS techniques. The interval of at least eight weeks from initial to follow-up training was too short for participants, who could not apply SMS between the two training sessions. A customized SNP training that is tailored to the specific needs of an oncology service in order to facilitate its effective long-term implementation should be considered. Supplementary training courses for healthcare professionals could play a crucial role in implementing SMS into daily routines.

A considerably lower number of health care professionals participated in the follow-up training (46 compared with 150 who participated in the initial training), which may have been due to limited personnel resources at the services, as reported by an external program evaluation [22]. In this evaluation, costs related to SNP training were not assessed or requested by healthcare administrators. Nonetheless, financial constraints may have functioned as an impediment, potentially contributing to the observed decline in participation rates from the initial to the follow-up training. Positive and negative statements after the follow-up training did not indicate that lack of interest was the reason for nonparticipation. The responses to the Work-SoC subdimensions suggest that participants perceived their work situation as *unpredictable* and *impossible to influence* rather than *predictable* and *easy to influence*. Only the Work-SoC score *manageable* changed significantly from initial to follow-up training. However, a notable association was not observed between participants’ perception of *manageability* and their empowerment to apply educational conversations post-initial training. This finding suggests that the SNP approach is well constructed and comprehensible to health care professionals but did not affect the participants’ work situation. Furthermore, our evaluation revealed a significant negative correlation between the perceived *manageability* and *comprehensibility* of the work situation and participants’ confidence in utilizing educational conversations after the follow-up training. This result aligns with the observation of a decline in confidence in applying educational conversations at follow-up training. Furthermore, the item ‘*manageable*’ exhibited a substantial decrease from the initial to the follow-up training. This decline could be related to the dynamic and varied nature of work environments in oncological care settings, which often necessitate adaptability to incorporate unanticipated responsibilities. The findings emphasize the notion that the work environment, characterized by overburdened workloads and/or staff shortages, may impede the implementation of educational conversations in clinical practice. Research has demonstrated that job resources impact on job performance [31,32,34,35,36]. The oncology working context of urban tertiary hospitals differs significantly from that of a rural oncology service with regard to staffing, infrastructure, organization, and nurse–physician collaboration [37]. Different working contexts pose challenges for the implementation of self-management interventions and should be given more consideration during the introduction and training. Education programs or tailored training on how to support self-management in practice are an important pillar, but they are not sufficient if the work context is perceived as a barrier to SMS [33].

Another barrier may be related to leadership. A survey of oncology nurses in Ontario, Canada showed that they ‘hold positive beliefs’ about SMS but were not always supported by senior managers [10]. Leadership commitment is important in providing staff with the time and resources to participate in SMS training such as the SNP. Supportive leadership can also make a difference in the context in which nurses work. Visible and relational leadership promotes the provision of quality care and creates a motivating work environment that can improve the quality of nursing care beyond the usual care [32]. Thus, supportive leadership may facilitate the integration of SMS into daily routines even when workloads are high. We did not ask about leadership in our evaluation. Future research should comprehensively assess the work context to inform potential systemic improvements to facilitate SMS in clinical routines. In addition, nursing advocacy in policy making is important in order to influence working conditions for oncology nurses. In Switzerland, the number of oncology nurses with an advanced nursing role (APN) is increasing [38]. APNs could be more involved in research [39] and help oncology health care professionals to implement complex interventions such as educational conversations with SN leaflets [38,40].

Among the strengths of this evaluation was first that the SNP has largely been implemented in Switzerland in different languages (i.e., German, French, and Italian). Evaluations of the implementation of a complex intervention training have rarely been reported but are key for future effectiveness studies. This evaluation is a first step in this direction. Second, the use of the Work-SoC scale provided some additional insight into how the training participants perceived their work context, which may help to understand why the implementation of a new intervention might not work. Studies using the Work-SoC scale have shown that work-related sense of coherence correlates with employee job satisfaction [23,36] and with oncology team resilience [29]. To the best of our knowledge, such detailed feedback on an SMS training has not been previously described. Therefore, further exploration of this topic is necessary to improve SMS implementation.

There are also a number of limitations. First, this evaluation was designed for quality assurance purposes. It is important to note that crucial information regarding the characteristics of the healthcare professionals such as participants’ age, gender, and educational background, was absent. Furthermore, the thematic analysis of the brief comments or key words employed was not a rigorous procedure. Therefore, our results should be considered with caution. To test the impact of the training on the implementation success, a proper pre- and post-test design would have been appropriate. Second, the context and conditions of cancer work differ not only between urban and rural areas but between countries [41]. Switzerland is a wealthy country with a highly ranked but expensive health care system [42]. This may affect the generalizability of our evaluation results to other SMS programs. Third, we evaluated the training content and methods, the participants’ confidence in having educational conversations, and their perceived work situation after each training session. This evaluation did not answer questions about the long-term implementation of the SNP. A third evaluation, conducted two months after the follow-up training, would provide valuable insights into the adoption of the SNP across various services. Fourth, we did not assess the barriers and facilitators affecting the adoption of the training methods. Future evaluations should also incorporate a cost-effectiveness analysis.

## 5. Conclusions

Participants confirmed that the SNP training content and methods largely met their expectations, although their confidence in using educational conversations decreased from the initial to the follow-up training after having applied the SNP in clinical routine. It appears that the *manageability* and *comprehensibility* of the work situation in cancer services correlated significantly negatively with the ability to integrate educational conversations into daily routines. The follow-up training will be adapted to provide more comprehensive support to healthcare professionals SMS skills, and complementary courses will be implemented to facilitate the application of educational conversations. Further research is needed to explore work-related context and SMS intervention costs to enable sustainable implementation of such programs.

## Figures and Tables

**Figure 1 curroncol-32-00326-f001:**
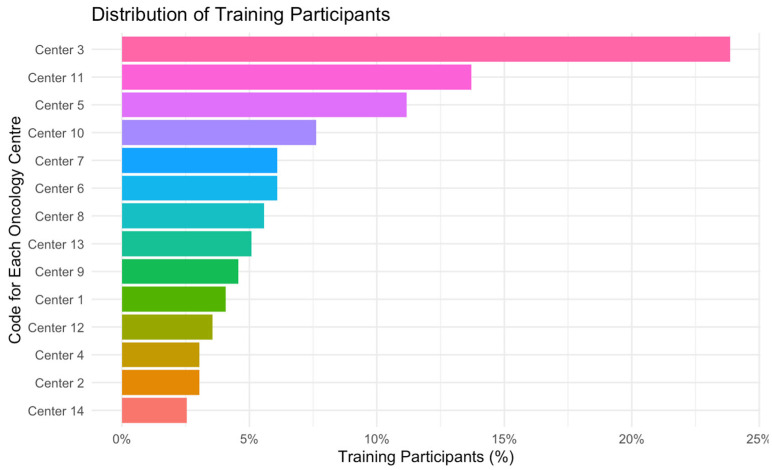
Distribution of SNP training participants at 14 centres. Legend: x axis, proportion of participants per centre including participants from both training sessions; y axis, code for each oncology centre.

**Table 1 curroncol-32-00326-t001:** Evaluation of the initial training (baseline).

Initial Training Items (Baseline)	N	Mean (SD)	Min/Max
Introduction was comprehensive Oral presentation was informative and comprehensible I learned from case studies I feel confident to apply educational conversations I feel confident to apply motivational interviewing	150 149 135 151 148	5.9 (0.9) 6.1 (0.8) 5.1 (1.6) 5.5 (1.3) 5.2 (1.4)	3/7 3/7 1/7 2/7 1/7

Likert scale 1–7 (not at all 1 2 3 4 5 6 7 very much).

**Table 2 curroncol-32-00326-t002:** Evaluation of the follow-up training (follow-up).

Follow-Up Training Items	N	Mean (SD)	Min/Max
I could ask my questions I got satisfied answers I feel empowered to apply educational conversations I feel confident to explain SNP leaflets I feel confident to apply educational conversations within my daily routines	45 45 46 46 46	5.9 (1.3) 6.1 (0.9) 5.5 (1.2) 5.8 (1.0) 4.8 (1.7)	1/7 3/7 2/7 3/7 1/7

Likert scale 1–7 (not at all 1 2 3 4 5 6 7 very much).

**Table 3 curroncol-32-00326-t003:** Comparison of Work-SoC scales between baseline and follow-up.

Work-SoC Items		Baseline			Follow-Up		
	N	Mean (SD)	Min/Max	N	Mean (SD)	Min/Max	*p*
**Comprehensibility** - structured - clear - manageable **Meaningfulness** - rewarding - significant - meaningful **Manageability** - easy to influence - controllable - predictable	128 126 127 128 126 128 119 126 126	3.5 (1.5) 3.3 (1.5) 2.9 (1.6) 1.9 (1.1) 2.0 (1.1) 1.8 (1.3) 3.8 (1.4) 3.6 (1.4) 4.2 (1.5)	1/7 1/7 1/7 1/7 1/6 1/7 1/7 1/7 1/7	43 43 42 43 43 43 43 43 43	3.5 (1.6) 3.4 (1.6) 3.5 (1.7) 1.9 (0.8) 1.9 (1.0) 1.7 (0.8) 3.7 (1.5) 3.7 (1.4) 4.0 (1.4)	1/7 1/6 1/6 1/7 1/6 1/4 1/6 1/6 1/7	0.765 0.622 **0.035** 0.795 0.647 0.315 0.507 0.800 0.404

Work-SoC questionnaire: How do you personally find your current job and work situation in general? Answer options, for example: manageable 1 2 3 4 5 6 7 unmanageable. **Lower ratings are related to better outcomes, e.g., structured or clear**.

**Table 4 curroncol-32-00326-t004:** Correlation between the Work-SoC subdimensions of comprehensibility and manageability and feeling empowered and confident to apply educational conversations after follow-up training.

Work-SoC Subdimensions	Comprehensibility		Manageability	
**Follow-Up Training Items**	**ρ\rhoρ**	**p**	**ρ\rhoρ**	* **p** *
I feel empowered to apply educational conversations	−0.2134	0.1748	−0.4295	**0.004535**
I feel confident to apply educational conversations within my daily routines	−0.3182	**0.03999**	−0.4155	**0.006208**

In bold are statistically significant *p* values.

**Table 5 curroncol-32-00326-t005:** Examples for feedback quotes for baseline and follow-up training.

What Was Particularly Positive About the Training	What Was Inappropriate at the Training
“The SNP is clearly and comprehensibly structured, topics well covered” (initial training, *comprehensibility,* centre 5) “The concept can be adapted for our setting” (initial training, *manageability*, centre 4) “Clarifying questions, exchanging ideas with colleagues” (follow-up training, *comprehensibility*, centre 5) “Very open exchange, questions can be clarified” (follow-up training, *comprehensibility*, centre 11) “The idea of coaching, away from just advising” (follow-up training, *meaningfulness*, centre 10) “We were encouraged that to implement the SNP is a feasible process” (follow-up training, *comprehensibility*, centre 7)	“Lots of theory, more opportunities to practice” (initial training, *meaningfulness*, centre 5) “Motivational interviewing was imprecisely explained/in too little detail for me to be able to implement it” (initial training, *comprehensibility*, centre 3) “No idea how to find the time to do it” (initial training, *manageability*, centre 9) “The timing was not appropriate for me as I could not apply educational conversations” (follow-up training, *manageability*, centre 6).

## Data Availability

The data can be obtained from M.B. upon request.

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
