# Peer review of "Self-Management Support for Cancer Survivors: A Descriptive Evaluation of the Symptom Navi Training from the Perspective of Health Care Professionals"

_curroncol, 2025, doi:10.3390/curroncol32060326_

Round 1
Reviewer 1 Report
Comments and Suggestions for Authors
Thanks for your invitation to review this manuscript. The SNP training content and methods met participants' needs, although participants' confidence in using the leaflets in educational conversations decreased after working with them. It appears that participants' perceptions of the comprehensibility and manageability of their work situation have an impact on whether SMS interventions can actually be implemented in clinical practice. While the study provides insightful analysis on the efficacy of the SNP for cancer survivors, several areas could be enhanced for greater impact. First, would you please exppand the discussion about the cost-effectiveness of the training program especially for healthcare administrators and policymakers? Second, are there any barriers and facilitators affecting the adoption of the training methods in real-world clinical practice? Third, is there any long-term follow-up to assess the sustainability of the training effects?
Author Response
Thank you very much for taking the time to review this manuscript. We comprehensively revised the manuscript based on the comments and inputs of four reviewers. Please find the detailed responses to your comments below and the corresponding revisions/corrections highlighted in the submitted files.
Comments 1: First, would you please expand the discussion about the cost-effectiveness of the training program especially for healthcare administrators and policymakers?
Response 1:
Thank you for this comment. We expanded the discussion with some reflections regarding costs. However, it is not possible to integrate statements on cost-effectiveness because we did not assess cost-related variables to conduct a cost-effectiveness analysis.
See p. 11, lines 397-400, p. 12. Lines 470-471, line 481
Comments 2: Second, are there any barriers and facilitators affecting the adoption of the training methods in real-world clinical practice?
Response 2:
Thank you for this question. Our submitted article describes the training evaluation in terms of content, methods, and perceived work situation from health care professionals’ perspective in real-world clinical practice. We did not integrate specific questions related to barriers and facilitators affecting the adoption of the training methods. However, our results show that the perceived work situation may be a potential barrier. Due to this shortcoming, we added as a fourth limitation on p. 12, lines 469-471 that facilitators and barriers were not specifically addressed in the evaluation.
Comments 3: Third, is there any long-term follow-up to assess the sustainability of the training effects?
Response 3:
We agree that long-term follow-up assessing the sustainability of the SNP training is important. This is included in the paragraph on limitations, p. 12, lines 467-469

Reviewer 2 Report
Comments and Suggestions for Authors
Dear authors,
thank you for your paper which presents an interesting and important approach in supporting cancer patients. I have noticed some ambiguities that should be clarified to make the study results easier to understand. My suggestions and comments are as follows:
Introduction
p. 1, line 40: What sort of collaboration or initiative is the Global Partners for Self-Management in Cancer? Please add some brief information, as not every reader will be familiar with them.
p. 2, line 51: Do the "different approaches to SMS" refer to training, treatment, educational... approaches aimed at cancer patients? Please outline briefly.
p. 2, line 79: Please write "salutogenetic" instead of "salutogeneic".
I would find it helpful if you could explain in a little more detail on why you chose work-related sense of coherence as a central construct. What was the rationale for this? Did you also consider other (work-related) variables? Regarding the hypotheses of the study (p. 4), one might also have considered parameters such as self-efficacy. A more detailed explanation of why work-related SoC was chosen could also contribute to a better understanding of the hypotheses in the methods section.
Methods
p. 3, line 100: Please write "design" instead of "desing".
p. 3, line 117: Did I understand it correctly that the institutions mentioned here correspond to the 13 facilities mentioned on page 2, lines 91/92? Please explain briefly.
p. 4, line 153: What was the reason for not collecting sociodemographic information from the participants?
p. 4, lines 180 et seq.: As mentioned above, it would be helpful to explain the focus on work-related SoC - and thus why the hypotheses were formulated as they are now. I.e., why did you assume associations between SoC meaningfulness and the confidence items, for example? This is not clear. How were the hypotheses derived?
p. 4, lines 196/197: This statement means that you used the work-related SoC items as categories for the qualitative data, right? If so, it might be better to write "subdimensions", "subdomains" or similar instead of "subscales" because the latter implies an analysis of quantitative data (in terms of questionnaire data).
Results
p. 5, line 218: Did you also conduct a confirmatory factor analysis, given the goodness of fit indices that are mentioned here? In any case, the specific values of TLI, RMSR etc should be listed in the text.
p. 6, figure 2: Please add what kind of coefficient the values in the figure represent.
p. 6, line 233: Does a mean of 4.8 really indicate that the participants felt not very confident to apply educational conversations? I would guess that this rather indicates a medium level of confidence.
p. 6, line 241: Please correct the reference to the table, it should be "table 4" instead of "table 3".
p. 7, hypotheses 1 and 2: I would suggest adding a sentence here on whether the hypotheses could be confirmed or not, this would be helpful for the reader.
Discussion
p. 9, lines 325/326: In the results section, you write that lower SoC scores indicate a higher/"better" level of the SoC constructs. That means in my understanding that the negative correlation between SoC scores and participants' skills that was found indicates "a higher sense of coherence is associated with higher subjective skills".
However, in the discussion section (as well as in the conclusion), you write that there is a "negative correlation between the perceived manageability and comprehensibility of the work situation and participants’ confidence in utilizing educational conversations after the follow-up training". That does not match and seems confusing. Please clarify.
p. 10: I would find it interesting to hear about the planned next steps in terms of future research regarding the evaluation and implementation of the intervention. Perhaps you could add a sentence or two about this?
Author Response
Thank you very much for taking the time to review this manuscript. We comprehensively revised the manuscript based on the comments and inputs of four reviewers. Please find the detailed responses to your comments below and the corresponding revisions/corrections highlighted in the submitted files.
Comments 1: Introduction
- line 40: What sort of collaboration or initiative is the Global Partners for Self-Management in Cancer? Please add some brief information, as not every reader will be familiar with them.
- line 51: Do the "different approaches to SMS" refer to training, treatment, educational... approaches aimed at cancer patients? Please outline briefly.
- line 79: Please write "salutogenetic" instead of "salutogeneic".
- I would find it helpful if you could explain in a little more detail on why you chose work-related sense of coherence as a central construct. What was the rationale for this? Did you also consider other (work-related) variables? Regarding the hypotheses of the study (p. 4), one might also have considered parameters such as self-efficacy. A more detailed explanation of why work-related SoC was chosen could also contribute to a better understanding of the hypotheses in the methods section
Response 1: Thank you for your comments related to our introduction. We have modified this part according to several reviewer comments. According to your points we changed the following texts:
1) we added on p.1, line 42-43) Therefore, the Global Partners for Self-Management in Cancer, an international collaboration of researchers, have …
2) Thank you for this comment: we clarified the different approaches on p.2, lines 54-55 with: “There is a growing body of evidence on the different approaches to SMS content, type, theoretical underpinning, and impact that have been introduced and tested in people with cancer, with varying effects on outcomes [6-9].”
3) Thank you for this correction: we changed the word accordingly to salutogenetic (p. 2, line 91)
4) We agree that more clarification why we chose the different theoretical underpinnings. We added more information on p. 2, lines 71-76, lines 78-81, and lines 94-95:
Comments 2: Methods
1. p. 3, line 100: Please write "design" instead of "desing".
2. p. 3, line 117: Did I understand it correctly that the institutions mentioned here correspond to the 13 facilities mentioned on page 2, lines 91/92? Please explain briefly.
3. p. 4, line 153: What was the reason for not collecting sociodemographic information from the participants?
4. p. 4, lines 180 et seq.: As mentioned above, it would be helpful to explain the focus on work-related SoC - and thus why the hypotheses were formulated as they are now. I.e., why did you assume associations between SoC meaningfulness and the confidence items, for example? This is not clear. How were the hypotheses derived?
5. p. 4, lines 196/197: This statement means that you used the work-related SoC items as categories for the qualitative data, right? If so, it might be better to write "subdimensions", "subdomains" or similar instead of "subscales" because the latter implies an analysis of quantitative data (in terms of questionnaire data).
Response 2: Thank you for pointing out several important questions regarding the Methods. We have, accordingly, revised the Method chapter to emphasize your mentioned points.
1) We corrected the word design on p.4, line 121.
2) Yes, the facilities correspond to the 14 facilities mentioned in the introduction (13 was a mistake we corrected according to the real number). We clarified this information on p. 4, line 125-127.
3) Thank you for raising this question. We added a rationale for not assessing socio-demographic data on p. 4, lines 141-147.
4) Thank you for this comment. We integrated additional information on choosing the Work-SoC scale in the Introduction part. See our response 1, 4).
We agree and changed the word to sub-dimensions (p. 6, line 242).
Comments 3: Results
1. p. 5, line 218: Did you also conduct a confirmatory factor analysis, given the goodness of fit indices that are mentioned here? In any case, the specific values of TLI, RMSR etc should be listed in the text.
2. p. 6, figure 2: Please add what kind of coefficient the values in the figure represent.
3. p. 6, line 233: Does a mean of 4.8 really indicate that the participants felt not very confident to apply educational conversations? I would guess that this rather indicates a medium level of confidence.
4. p. 6, line 241: Please correct the reference to the table, it should be "table 4" instead of "table 3".
5. p. 7, hypotheses 1 and 2: I would suggest adding a sentence here on whether the hypotheses could be confirmed or not, this would be helpful for the reader.
Response 3: Thank you for your comments on the results. We have, accordingly, modified several paragraphs to answer these points:
1. Confirmatory factor analysis was not conducted. This study was explicitly designed for exploratory scale clarification, so we limited the psychometric work to an EFA. Our reporting therefore focuses on the established EFA diagnostics - eigenvalues, scree and parallel analyses, item communalities and the rotated loading pattern. We plan to conduct a confirmatory factor analysis on a new, independent (and larger) sample in the next phase of the project. We decided to comprehensively revise the chapter Results and uploaded a supplementary file for the EFA.
2. The numbers in this Figure are standardized factor loadings obtained from the Principal Axis Factoring solution after Varimax rotation. This figure is included in the supplementary file as well.
3. Thank you for this comment. We agree and revised the text accordingly at p. 7, line 279.
4. Thank you for this comment. Due to a comprehensive manuscript revision, several table numbers changed.
5. Thank you for this comment. We introduced both hypothesis results with an interpretative sentence in this sense. P. 8, lines 299 and 303.
Comments 4: Discussion
p. 9, lines 325/326: In the results section, you write that lower SoC scores indicate a higher/"better" level of the SoC constructs. That means in my understanding that the negative correlation between SoC scores and participants' skills that was found indicates "a higher sense of coherence is associated with higher subjective skills".
However, in the discussion section (as well as in the conclusion), you write that there is a "negative correlation between the perceived manageability and comprehensibility of the work situation and participants’ confidence in utilizing educational conversations after the follow-up training". That does not match and seems confusing. Please clarify.
p. 10: I would find it interesting to hear about the planned next steps in terms of future research regarding the evaluation and implementation of the intervention. Perhaps you could add a sentence or two about this?
Response 4:
Thank you for raising this issue. We discussed the results with the statistician and co-author Selma Riedo. Higher Work-SoC scores must be negatively interpreted: i.e., for manageability difficult to influence, uncontrollable, and unpredictable. Such negative sense of coherence is associated with lower subjective skills. We therefore clarified this interpretation on p. 9, lines 310-312.
We added some information regarding planned next steps (adaptation of the follow-up training and complementary courses) on p. 12, lines 478-480.

Reviewer 3 Report
Comments and Suggestions for Authors
In this article, the authors describe evaluation of a self-management support training for cancer survivors. Registered nurses, psychologists and social workers were included as participants of the evaluation. To this end, Work-SoC questionnaires were used as well as qualitative assessment by narrative feedback. Two rounds of training interventions were performed with 141 participants in the first round, and 44 in the second one. Except for an increase in the item “manageable", there were no significant differences in between the answers of the first vs. the second training round. The authors conclude that manageability and comprehensibility of the work situation in cancer services correlates significantly negatively with the ability to integrate educational conversations into daily routines. The paper also provides some insight in the perceptions of health care providers that go beyond the immediate evaluation of the training, e.g., on the work environment.
While self-management support for cancer survivors is highly relevant as is the evaluation of the effectiveness of the methods used, I have, however, some concerns that should be addressed:
- The title and abstract of this article are somewhat misleading. From the title of the paper one might expect that the evaluation of a self-management support for cancer survivors would actually deal with cancer patients' perspective. Moreover, there is no clear description contained in the abstract of who the participants of the study were.
- The authors report that the Symptom Navi Program (SNP) was developed more than a decade, and there are two references of published papers. Nonetheless, it would be helpful and informative for the readers if some information on the content would be provided in the introduction instead of merely a description of the materials used (leaflets, nurse-led semi-structured educational consultations and standardized SNP implementation training and a manual for health care providers). In addition, such information could be provided as supplemental files or hyperlinks to generally accessible resources.
- It is stated that registered nurses, psychologists, and social workers from 14 centers who attended at least one SNP training session were included, but no information on the recruitment or inclusion criteria, respectively, of these participants is provided. How were the respective professionals chosen or referred to the training program? Was this open to any health care professional, or had they been selected by the organizers of the Navi program due to job experience, special expertise or other prerequisites?
- No statistically significant difference was observed between baseline and follow-up Work-SoC scores. Hence, a positive development or improvement, respectively, during the course of training is questionable. Since this issue is critical with regard to the effectiveness of the intervention (or lack thereof), it should be discussed more in depth.
- Indeed, there is a striking reduction of participants in the second round of training (decrease from 141 participants in the first round and only 44 in the second one). Have all participants completed the questionnaires, or are there missing data in the first or second round?
- Are the answers similar for all different centers, or are there any relevant differences? For instance, differences between the perception of healthcare providers in clinical vs. outpatient centers could be interest. In addition, are there noticeable differences between the perspective of nurses vs. psychologists?
- It would be of utmost interest to understand the negative motivation of those participants of the first round of training who did not join the second round. Have they actually been contacted, and how did they explain their lack of interest to continue. The authors assume that this was due to limited personnel resources at the services. On the other hand, the absence of more than two third of participants in the second round of training could well be a confounding factor since it cannot be excluded that they did not return because of a more negative perception of the experience of the training.
- The term “significance" should only be referred to if levels of significance can be attributed whereas different expressions should be preferred for meaningfulness. (“A significant lower number of health care professionals participated in the follow-up")
- The study design that does not contain informations on the participants such as age, gender, education or health-related information should be discussed as a limitation. In addition, since respondents were invited to offer brief comments or key words in response to the questions, a stringent analysis of the data is precluded, which should be included as another limitation.
Minor criticism:
All abbreviations should be written out by the first time of use in the text (e.g., International Society of Nurses in Cancer Care for ISNCC).
Comments on the Quality of English LanguageThe English is generally o.k. but some English expressions might be improved by editing.
Author Response
Thank you very much for taking the time to review this manuscript. We comprehensively revised the manuscript based on the comments and inputs of four reviewers. Please find the detailed responses to your comments below and the corresponding revisions/corrections highlighted in the submitted files.
Comments 1: The title and abstract of this article are somewhat misleading. From the title of the paper one might expect that the evaluation of a self-management support for cancer survivors would actually deal with cancer patients' perspective. Moreover, there is no clear description contained in the abstract of who the participants of the study were.
Response 1:
We agree with your comment and adapted the title to
Self-management support for cancer survivors: a descriptive evaluation of the Symptom Navi training from the perspective of health care professionals
We completed the abstract with information on who were the participants of the program evaluation, p. 1, line 18-19 .
Comments 2: The authors report that the Symptom Navi Program (SNP) was developed more than a decade, and there are two references of published papers. Nonetheless, it would be helpful and informative for the readers if some information on the content would be provided in the introduction instead of merely a description of the materials used (leaflets, nurse-led semi-structured educational consultations and standardized SNP implementation training and a manual for health care providers). In addition, such information could be provided as supplemental files or hyperlinks to generally accessible resources.
Response 2:
Thank you for pointing this out. We agree with your comment and have replaced information on training theoretical underpinning (COM-B model, Michie et al., 2011) from chapter 2.2 into the introduction at page 2, lines 71 to 76. Additional information related to training content is integrated in the Methods chapter, p. 4., lines 157-160.
We considered your suggestion to upload supplementary files or hyperlinks to generally accessible resources. SNP resources are available in German and French and therefore may not be understood by international readers. We therefore decided to include the information into the manuscript.
Comments 3: It is stated that registered nurses, psychologists, and social workers from 14 centers who attended at least one SNP training session were included, but no information on the recruitment or inclusion criteria, respectively, of these participants is provided. How were the respective professionals chosen or referred to the training program? Was this open to any health care professional, or had they been selected by the organizers of the Navi program due to job experience, special expertise or other prerequisites?
Response 3:
Thank you for this comment. We agree that this information was not clear and have modified the corresponding paragraph to add more information on training participants on p. 4, line 125-127. On p. 4, lines 157-160, we added the information that the cancer services decided who from their team participated in the training sessions.
Comments 4: No statistically significant difference was observed between baseline and follow-up Work-SoC scores. Hence, a positive development or improvement, respectively, during the course of training is questionable. Since this issue is critical with regard to the effectiveness of the intervention (or lack thereof), it should be discussed more in depth.
Response 4:
Thank you for raising this point. We assessed the work situation from training participants perspective. The aim of the SNP training was to support health care professionals integrating SMS into their daily routines. We agree with your comment that this issue is important and added more reflections in the Discussion on p. 11, lines 401-402, lines 406, and lines 414-416.
Comments 5: Indeed, there is a striking reduction of participants in the second round of training (decrease from 141 participants in the first round and only 44 in the second one). Have all participants completed the questionnaires, or are there missing data in the first or second round?
Response 5:
Thank you for pointing out this question. We have, accordingly, added more information on return rate to emphasize this point at p. 7, lines 254-263.
Comments 6: Are the answers similar for all different centers, or are there any relevant differences? For instance, differences between the perception of healthcare providers in clinical vs. outpatient centers could be interest. In addition, are there noticeable differences between the perspective of nurses vs. psychologists?
Response 6:
Thank you for this comment. Because we did not assess participants’ personal information, we cannot make such interpretations on different answers between healthcare providers. We integrated this information on p. 10 (Discussion) and added a statement based on oral feedback we got during the training sessions, lines 378-382.
Comments 7: It would be of utmost interest to understand the negative motivation of those participants of the first round of training who did not join the second round. Have they actually been contacted, and how did they explain their lack of interest to continue. The authors assume that this was due to limited personnel resources at the services. On the other hand, the absence of more than two third of participants in the second round of training could well be a confounding factor since it cannot be excluded that they did not return because of a more negative perception of the experience of the training.
Response 7:
Thank you for raising this point. The oncology services organized the training sessions at their center and decided who of their team will participate the training sessions. We added information to better explain the decrease of participants at the follow-up trainings on p. 7, lines 254-263 (to respect the interval between initial and follow-up training for 2 centers, 2 centers decided not to organize a follow-up training, and from one center we did not receive the completed questionnaires after the follow-up training). This argument is discussed on p. 11, lines 396-401.
Comments 8: The term “significance" should only be referred to if levels of significance can be attributed whereas different expressions should be preferred for meaningfulness. (“A significant lower number of health care professionals participated in the follow-up")
Response 8:
Thank you for this point. We replaced the word significant by considerably lower number on p. 11, line 393.
Comments 9: The study design that does not contain informations on the participants such as age, gender, education or health-related information should be discussed as a limitation. In addition, since respondents were invited to offer brief comments or key words in response to the questions, a stringent analysis of the data is precluded, which should be included as another limitation.
Response 9:
We agree with your comment and added this information to the limitations on p. 12, lines 455-457.
Comments 10: Minor criticism:
All abbreviations should be written out by the first time of use in the text (e.g., International Society of Nurses in Cancer Care for ISNCC).
Response 10:
Thank you for this comment. We added International Society of Nurses in Cancer Care for ISNCC on p. 10, line 362.
Point 1: The English is generally o.k. but some English expressions might be improved by editing.
We edited the whole document and improved / modified English expressions.

Reviewer 4 Report
Comments and Suggestions for Authors
General comments to the authors
The topic of the manuscript, self-management support in cancer is of interest in the current health care scenario. Below, I provided recommendations to address certain features of the manuscript that require attention before this piece is recommended for publication.
The whole manuscript will benefit from proofreading as there are some typos along the document (e.g., line 100 should read ‘design’ as instead of ‘desing’).
Specific comments to the authors
- Abstract. Some key information is missing in this section. Please address the following:
- What does this exactly mean: “An initial and a follow-up SNP training”? Does this refer to a preliminary try? A pilot-testing initiative?
- What was the rationale and the objective of this inquiry?
- What type of study is this?
- Introduction. The presentation of the problem has been properly done. It would be interesting to have a more specific information of:
- Whether the present piece is a research study or an evaluation of a programme
- the difference between the pilot study and the first objective of the present inquiry
- As per the item 2.2, the present intervention was scaled up. This needs to be stated in this section and the framework and strategies used to achieve this purpose should be presented as the evaluation has to taken them into account (Michie et el., 2011)
- Methods. Some issues require attention:
- Regarding the evaluation design sub-section, please see the following:
- The first sentence reads: “This exploratory evaluation assessed the SNP training content, methods, participants confidence to use educational conversations in daily routine, as well as participants perception of their individual work situation at two time points”. My question is: if participants’ perceptions were used to assess “individual work situation at two time points”, how were “the SNP training content, methods, participants confidence to use educational conversations in daily routine” assessed? Was it not using the participants perspectives as well?
- if this is an exploratory evaluation, why are descriptive questions being asked (i.e., ‘how…’)?
- The evaluation was conducted in accordance with the Declaration of Helsinki, the principles of Good Clinical Practice, and the Swiss Human Research Ordinance; then, what this managed?
- Nowhere is indicated that the present is a mixed design
- Comments related to the participants and SNP training procedures:
- The intervention was developed, pilot-tested, and scale-up. As all this happened before the evaluation, I think it is pertinent to move this information to the introduction as being here brings the impression that this was done for the purposes of this evaluation
- The training programme is based on the COM-B model and included the coaching approach based on the 5 A’s 134 (assess, advise, agree, assist, arrange), as well as the principles of motivational interviewing. Please explain how these three models were coherently integrated. Also, the training model has some features, why have these features been used (and not others)? Rationale is needed.
- Data analysis: Two points. Have data been integrated? Yes? No? Why? How? Also, I would argue that the thematic analysis described in this section does not transpire in the results in any sense.
- Regarding the evaluation design sub-section, please see the following:
- Results. In general, I found this section somehow hard to follow and disconnected from the methodology.
- Discussion. Overall, this section reads well but it will have to be reviewed to align to the proposed changes indicated above.
Author Response
Thank you very much for taking the time to review this manuscript. We comprehensively revised the manuscript based on the comments and inputs of four reviewers. Please find the detailed responses to your comments below and the corresponding revisions/corrections highlighted in the submitted files.
Comments 1: Abstract. Some key information is missing in this section. Please address the following:
- What does this exactly mean: “An initial and a follow-up SNP training”? Does this refer to a preliminary try? A pilot-testing initiative?
- What was the rationale and the objective of this inquiry?
- What type of study is this?
Response 1: Thank you for pointing this out. We see that some information is missing and have revised the abstract by including information on the rationale (assessing training quality) related to a scale up dissemination from 2021 – 2024. To respect abstract word count limitation, it is not possible to give details on SNP training. However, we added this information in the introduction.
Comments 2: Introduction. The presentation of the problem has been properly done. It would be interesting to have a more specific information of:
- Whether the present piece is a research study or an evaluation of a programme
- the difference between the pilot study and the first objective of the present inquiry
- As per the item 2.2, the present intervention was scaled up. This needs to be stated in this section and the framework and strategies used to achieve this purpose should be presented as the evaluation has to taken them into account (Michie et el., 2011)
Response 2:
Thank you for your valuable suggestion. We have accordingly modified several paragraphs of the introduction to your comments:
1) Our manuscript reports on a program evaluation; we added information for clarification on page 3, line 113, and p. 4, line 122.
2) We added information on p. 3, lines 110-113 (“Scaling-up the use of SNP across different oncology services and health care professionals requires ongoing quality assessment. We therefore extended the evaluation of each SNP training with the same evaluation tools as in the pilot study” and..”to complement pilot study data; .”)
3) we added information on scaling up the SNP on p. 3, lines 110-112; To clarify information related to framework and strategies, we added / replaced information into the Introduction. See p. 2, lines 71-76.
Comments 3: Methods. Some issues require attention:
Regarding the evaluation design sub-section, please see the following:
- The first sentence reads: “This exploratory evaluation assessed the SNP training content, methods, participants confidence to use educational conversations in daily routine, as well as participants perception of their individual work situation at two time points”. My question is: if participants’ perceptions were used to assess “individual work situation at two time points”, how were “the SNP training content, methods, participants confidence to use educational conversations in daily routine” assessed? Was it not using the participants perspectives as well?
- if this is an exploratory evaluation, why are descriptive questions being asked (i.e., ‘how…’)?
- The evaluation was conducted in accordance with the Declaration of Helsinki, the principles of Good Clinical Practice, and the Swiss Human Research Ordinance; then, what this managed?
- Nowhere is indicated that the present is a mixed design
Response 3:
Thank you for your comments on the Methods. Based on several comments from all reviewers, we comprehensively revised the Methods. We agree that our article was not clear on design characteristics and ethical considerations. We have, accordingly, revised the text on page 4 and 5 to emphasize these points:
- We confirm that the whole program evaluation is based on participants perspectives. We modified the first sentence on p. 4, line 122
- The word ‘explorative’ has been changed to descriptive, p. 4, line 122
- We added further information on p. 4, lines 149-151
- We have deliberately not labelled a research design, as our article reports a quality program evaluation. For this reason, we do not wish to characterize our descriptive evaluation as a mixed-method design.
Comments 4: Comments related to the participants and SNP training procedures:
- The intervention was developed, pilot-tested, and scale-up. As all this happened before the evaluation, I think it is pertinent to move this information to the introduction as being here brings the impression that this was done for the purposes of this evaluation
- The training programme is based on the COM-B model and included the coaching approach based on the 5 A’s 134 (assess, advise, agree, assist, arrange), as well as the principles of motivational interviewing. Please explain how these three models were coherently integrated. Also, the training model has some features, why have these features been used (and not others)? Rationale is needed.
Response 4:
We agree with your comments that information related to SNP pilot-testing and scale up, as well as on theoretical underpinnings should be replaced to the Introduction. We accordingly replaced and completed this information on p. 2, lines 71-76 and lines 78-81. The rationale for the COM-B model is added: p. 2, line 71-72.
Comments 5: Data analysis: Two points. Have data been integrated? Yes? No? Why? How? Also, I would argue that the thematic analysis described in this section does not transpire in the results in any sense.
Response 5:
We understand that your question is related to a mixed-method design. We conducted a program evaluation. We agree that this was not clear in the Methods and added a corresponding statement at p. 6, lines 244-245.
We agree with your argument that the thematic analysis should be improved in the Results. We revised the Results in our manuscript comprehensively and emphasized in more details the qualitative results on p. 9, lines 315-337. Thematic Analysis phases 1 to 3 are described in text, and phases 4 and 5 in table 5 on p. 9.
Comments 6: Results. In general, I found this section somehow hard to follow and disconnected from the methodology.
Response 6:
As mentioned above, we revised the whole chapter on the Results comprehensively to address your comment:
- we decided to move the results on the Exploratory Factor Analysis in a supplementary file, as it is relevant with respect to the scale properties rather than for the program evaluation.
- we included additional information on the participating oncology services and participants (p. 7, lines 247-248, and lines 255-264)
- we revised the results on Work-SoC hypothesis testing (p. 8 and 9, lines 299-316)
Comments 7: Discussion. Overall, this section reads well but it will have to be reviewed to align to the proposed changes indicated above.
Response 7:
After the revision of the results section, we modified several paragraphs in the discussion on p. 10 (lines 343-346, 352, 354-355, 357-359, 370-371, 376-390), p.11 (lines 391-393, 397-402, 404-405, 408-409,417-419), and p. 12 (lines 452-454, 456-458, 467-471).
The English is fine and does not require any improvement.
The whole manuscript will benefit from proofreading as there are some typos along the document (e.g., line 100 should read ‘design’ as instead of ‘desing’).
Response to comment on English language:
We carefully checked for typos along the document and conducted a proofreading before submitting the revised manuscript.

Round 2
Reviewer 2 Report
Comments and Suggestions for Authors
Dear authors,
thank you for the thorough and comprehensive revision of your manuscript which is now, in my view, better readable and understandable. My suggestions were adequately considered. I would like to point out two remaining aspects:
p. 9, line 311: It would be helpful to add in that sentence that higher SoC scores imply a lower sense of coherence, e.g.:
"The negative correlation suggests that higher Work-SoC sub-dimension scores (i.e., a lower sense of coherence) are associated with a decrease of feeling empowered or confident to apply educational conversations."
And one comment on the factor analysis:
You have now added a supplementary file with the results of the EFA. There, you mention a good model fit (RMSR, TLI) but the values for these fit indices are lacking. Please add them to the file. Moreover, it would be good to list the values for the EFA mentioned in the file (eigenvalues, screeplot) for the sake of completeness, since you have created this supplement specifically.
Author Response
Dear Reviewer,
Thank you for taking the time to review our manuscript a second time. We agree with your comments and have carefully revised our manuscript accordingly. The page and line references refer to the uploaded document in track-change mode.
Sincerely,
The author team
I would like to point out two remaining aspects:
Comment 1) 9, line 311: It would be helpful to add in that sentence that higher SoC scores imply a lower sense of coherence, e.g.: "The negative correlation suggests that higher Work-SoC sub-dimension scores (i.e., a lower sense of coherence) are associated with a decrease of feeling empowered or confident to apply educational conversations."
Thank you for this input. We have added this information to the text as suggested on p.9, line 334.
Comment 2) And one comment on the factor analysis:
You have now added a supplementary file with the results of the EFA. There, you mention a good model fit (RMSR, TLI) but the values for these fit indices are lacking. Please add them to the file. Moreover, it would be good to list the values for the EFA mentioned in the file (eigenvalues, screeplot) for the sake of completeness, since you have created this supplement specifically.
We added on the supplementary file the missing values for the EFA. The eigenvalues for the three extracted factors were 1.69, 1.62, and 1.42, respectively. The RMSR was 0.02 and the Tucker-Lewis Index (TLI) was 1.004, indicating good model fit.
Reviewer 4 Report
Comments and Suggestions for Authors
General comments to the authors
The review of the manuscript notably improved its quality. Few minor comments to address though.
The whole manuscript will benefit from proofreading as there are some typos along the document; this time, the structure of some sentences requires attention.
Specific comments to the authors
- Abstract. Ok
- Introduction. Ok
- Methods. Some issues still require attention:
- Regarding the evaluation design sub-section, please see the following:
- There is no reference provided for the evaluation framework used
- Nowhere is indicated that the present is a mixed design
- Comments related to the participants and SNP training procedures:
- Still, the theoretical integration in the development of the intervention has not been explained.
- Data analysis:
- I will keep on arguing the authors did not conduct reflective thematic analysis as this is not transpired in this section, nor in the results
- Also, were the quantitative and qualitative analysis conducted consecutively or concurrently?
- Results. The section reads well.
- Discussion. Overall, this section reads well; but:
- The first paragraph is too heavy and its purpose it watered
- What do you mean by “narrative feedback”?
- Regarding the evaluation design sub-section, please see the following:
Mentioned above
Author Response
Dear Reviewer,
Thank you for taking the time to review our manuscript a second time. We agree with your comments and have carefully revised our manuscript accordingly. The page and line references refer to the uploaded document in track-change mode.
Sincerely,
The author team
Comment 1) The whole manuscript will benefit from proofreading as there are some typos along the document; this time, the structure of some sentences requires attention.
Response 1): Thank you for this comment. We have read the entire manuscript and improved the English language. We make this visible with the “Track Changes” mode. Consequently, two versions of the document are uploaded: a clean document and a document in track change with the adapted passages from the first round highlighted in yellow.
Comments 2) Methods. Some issues still require attention:
a) Regarding the evaluation design sub-section, please see the following:
- There is no reference provided for the evaluation framework used / Response: Thank you for this comment. We added “We based the evaluation methods on the Medical Research Council guidelines for the development and evaluation of complex interventions [21].” P. 4, lines 127-129.
- Nowhere is indicated that the present is a mixed design. / Response: You raised this point in your comments on the first round. We argued then that because we were reporting a quality program evaluation, we didn't want to use a scientific research design to label our evaluation. In response to your argument, we added: "Both quantitative and qualitative methods were used." On p. 4, line 127.
b) Comments related to the participants and SNP training procedures:
- Still, the theoretical integration in the development of the intervention has not been explained. Response: We agree that this information is usually included in the methods. Responding to other reviewers’ input at round 1, we added information on SNP training procedures, the rational and theoretical frameworks in the introduction. The introduction is more appropriate, as the development of the intervention development took place before the start of the descriptive program evaluation reported in our manuscript. Please see p. 2, lines 71-76, 79-82, and 95-96.
c) Data analysis:
- I will keep on arguing the authors did not conduct reflective thematic analysis as this is not transpired in this section, nor in the results. Response: We understand that the results based on thematic analysis were less detailed reported than the quantitative results. Nevertheless, we did follow the six phases described by Braun and Clark and integrated these findings in the report (phase 6). We added information on p.6, lines 256-263 to make this clearer. Further we added a paragraph in the results to emphasize the thematic analysis approach. P. 9, lines 342-349.
- Also, were the quantitative and qualitative analysis conducted consecutively or concurrently? Response: Consecutively: This is written explicitly on p. 6, line 264.
d) Discussion. Overall, this section reads well; but:
- The first paragraph is too heavy and its purpose it watered. Response: Thank you for this comment. We have divided the first paragraph, improved its structure, and moved some statements to make it easier to understand.
- What do you mean by “narrative feedback”? Response: “Narrative feedback” refers to responses to open-ended questions written as keywords or short narrative descriptions. We have added an explicit explanation on p. 6, line 253, and replaced the word feedback with statements in the manuscript.